# Association of Academic Stress, Acne Symptoms and Other Physical Symptoms in Medical Students of King Khalid University

**DOI:** 10.3390/ijerph19148725

**Published:** 2022-07-18

**Authors:** Farah Aziz, Mohammad Fareed Khan

**Affiliations:** 1Department of Basic Medical Sciences, College of Applied Medical Sciences, King Khalid University, Khamis Mushait 62529, Saudi Arabia; 2Department of Laboratory Medicine, Abha International Private Hospital, Abha 61431, Saudi Arabia; fareedkhandurrani@gmail.com

**Keywords:** academic stress, acne, physical symptoms, mental health, medical college students

## Abstract

Academic stress has varied effects on medical student life due to multiple factors, like study load, competition, frequent assessment, social pressure, etc. The authors of this paper conducted research to find the academic stress level and its sequel over acne and other physical symptoms on the medical students of King Khalid University (KKU), Saudi Arabia. A total of 168 participant responses were analyzed. Data collection was performed using a self-administered online questionnaire through the university website portal. The study tool was comprised of four sections: demographic characteristics, academic stress, acne symptoms, and other physical symptoms. Statistical analysis was performed using SPSS software. A high proportion of females (88.7%) participated in the study. Upon categorization of overall academic stress, it was found that a majority of the medical students were moderately stressed (58.34%). The response on the academic stress scale revealed that exams are the major cause of stress among students. The Mean ± SD of academic stress, acne symptoms, and physical symptoms differ significantly at <0.01 level of significance. Overall academic stress showed a significant positive association with acne (<0.01) and physical symptoms (<0.01). The strength of this study is the fact that its categorization of stress caused by academics has not been done elsewhere. In addition, the impact of acne and physical symptoms has not been found in recent literature. Keeping the outcome of the present study in mind, it is suggested to arrange timely counselling sessions in medical colleges which can alert medical students to remain conscious about the consequences of stress.

## 1. Introduction

Stress is a menace to an individual’s welfare and can be caused by various stressors like environmental, psychological, biological, and social factors. Stressful life events are followed by anxiety and depression. Long-term stress results in behavioral changes like smoking, drug addiction, sleeping problems, eating disorder, diminished appetite, and sedentary lifestyle [1]. Young adults experience problems such as a lack in evolvability, stress, and perplexity because of numerous stressors. Academic stress among students has been previously researched and several marked stressors have been identified, such as too many assignments, competition with other students, failures etc. An irresistible requirement to excel in studies leads to academic stress that can impair morale, harm cognition and learning, and lead to the breakdown of students [1,2].

Academic stress inflicts various patterns of stress response like breakouts on skin, acne, lack of energy, taking over the counter medication, high blood pressure, feeling depressed, increase in appetite, trouble concentrating, restlessness, tension, anxiety, and disturbed physical health [3]. Exposure to intense and chronic stressors during the developmental years has long-lasting neurobiological effects, including a high risk of anxiety, mood disorders, aggressive dyscontrol problems, hypoimmune dysfunction, medical morbidity, structural changes in the CNS, and early death [4]. The daily life aggravations become chronic and disturb individual health conditions. Intriguing results from studies depict the effect of chronic stress on pathophysiology of cardiovascular disease, upper respiratory tract infection, exacerbation of autoimmune disease, and cumulative increase in allostatic load [3,5].

Acne has a high worldwide prevalence rate of 85% in adolescents; stress is widely thought to trigger acne [6]. Persistent academic stress may have a role in acne exacerbation and physical symptoms. Research suggests that stress triggers the release of neuroactive substances within the epidermis that can activate inflammatory processes in the skin [7].

Besides the fact of stress and acne relation, only a few studies have been conducted on the impact of specific stressors, i.e., academic stress, on acne and other physical symptoms. The present study seeks to assess the level of academic stress and highlight its response on acne symptoms and other physical symptoms among medical students of KKU.

## 2. Materials and Methods

A cross-sectional study was conducted within KKU medical students of Abha and Khamis Mushait campuses. A convenience sampling method was adopted for data collection using an online questionnaire, distributed through university website portal. Participation in the study was voluntary and all the participants were of Saudi ethnicity. Students from study level three to internship were included in the study. Those with any skin-related disease, pregnancy, those on hormonal replacement therapy, and students of study level one and two (first and second semester) were excluded.

The sample size was calculated using Cochran’s formula with 5% accuracy level and 95% confidence interval, resulting a total of 168 studied participants.

The questionnaire was comprised of four sections containing closed-ended type questions. First section: Demographic characteristics like gender, age, self-reported height (cm), and weight (kg); place of the study; marital status (married or unmarried); study department (medical); and study level. Second section: the academic stress scale, originally developed by Kim (1970) and since utilized by numerous other researchers, was employed in this study [8]. It consisted of 14 items reporting several sources of study-related stress. Each item had five options with an assigned score i.e., none (0), slight stress (1), moderate stress (2), high stress (3), extreme stress (4). Thus, the maximum score was 56 for each factor. Depending upon the scores range, academic stress was categorized into low (0 to 18), medium (19 to 37), and high (38 to 56). Third section: Acne symptoms questions targeted to assess the physical symptoms of the facial acne sufferers. This section included five items. Each item was scored on a 0–6 point scale (extensive, a whole lot, a lot, a moderate amount, some, few, none). Maximum points for this domain ranged from 0 to 30 [9]. Fourth section: Other physical symptoms were assessed by 11 questions scored on dichotomous scale (yes = 1, no = 2). Possible physical symptoms score extended from 0 to 11 [10].

Data were analyzed using SPSS software version 25. Mean and standard deviation of independent variables; i.e., academic stress; dependent variable; i.e., acne symptoms; and other physical symptoms were calculated. ANOVA was applied at 0.01 level of significance among all the studied variables and different academic stress levels followed by Tukey’s honest test for pairwise comparisons between all groups. Pearson correlation was applied to observe the effect of overall academic stress on acne symptoms and other physical symptoms.

Ethical approval was obtained by an institutional ethical committee. An informed consent was stated and acquired from the participants before attempting the questionnaire.

## 3. Results

Table 1 shows a total of 168 medical student responses, within which a preponderance of females, with 88.7% of the total, was observed. Most participants were aged between 22–25 years (63.9%), were studying in Abha campus (66.3%), and were single (71.4%). Study level VII (seventh semester) had the fewest number of responses, while a large percentage of internship students (41.6%) participated in the study.

The scoring of academic stress scale evaluates the low, medium, and high stress levels among the participants as shown in Table 2. A majority of participants (*n* = 98) showed medium academic stress. Upon comparison, a significant difference was observed between all stress categories at 0.0001. A Post hoc test revealed the significant difference between individual groups at *p* < 0.01.

Table 3 depicts the Mean ± SD values of academic stress, acne symptoms, and other physical symptoms differing at <0.01 level of significance. Varied superscripts on the values showed the significant difference at <0.01, upon pairwise comparison with individual group. Correlating the Mean ± SD of academic stress with other physical symptoms revealed a stronger positive correlation (*p* < 0.00001) than acne symptoms (<0.01).

## 4. Discussion

Academic stress has been well noted among pre-university and university students [11]. Across the globe, studies carried out at universities in Thailand, Malaysia, the United Kingdom, and India observed academics as a perceived stressor among students [12,13,14,15,16,17,18]. Scientific literature indicates that stress is quite prevalent in Saudi academia, particularly in female students studying health-related courses [19,20,21]. In the present study, a majority of participants were females, belonged to medical departments, and were single. Medical education has been eminent and a foremost choice among students [19]. Consequently, medical programs are more competitive, challenging, and stressful in students. Sherina et al. and Zaid and Chan (2007) observed that the prevalence of stress among medical students ranged from 30% to 50% [22,23]. Abdel Rahman et al. assessed academics as a common stress agent among medical students [19]. In addition, Fatima et al. suggested frequent examinations and a difficult syllabus as the root cause of academic stress [24]. In the current study, all participants belonged to medical departments and taking academics as a stressor, a significant predominance of medium academic stress (58.3%) was observed over high and low academic stress. Stress levels on different genders were not assessed due to very few male participants. Therefore the strength of this study relates to academics as a particular stressor; stress categorization has not been done elsewhere. On the academic stress scale, a high stress response was observed towards examination-related questions, while cognitive and intellect-related questions moderately stressed the participants. Studies suggested that students fear that failure in examinations will leads to insults and social problems together, which in turn lead to perceived anxiety and depression, negative life-style practices, and a worse status of physical and mental health changes since the start of their college studies [25,26,27].

Numerous studies have revealed that stress plays a major role in acne exacerbation [28,29,30,31,32,33]. Chilicka K et al., reported female university students with acne possess negative affectivity and more social inhibition than female students without acne [34]. Moreover, intense and highly ambitious people reported more acne symptoms during frustration and stressful events [35,36]. In this study, the calculated correlation coefficient of 0.2 suggests a convincing association between overall academic stress and acne symptoms. Subjects who demonstrated the greatest increase in perceived stress during examination also displayed the greatest exacerbation of acne severity in a proportional predictable manner [7]. The data collection for this study was during the semester when regular lectures, exams, and other academic activities were at their peak, resulting stressed students observing acne symptoms. However, it may be possible that other factors like facial hygiene, menstrual cycle, hormonal influence, and unhealthy diet may be the reason of acne symptoms in females. Theories of dermatology suggest that stress-induced neuroactive substances within the epidermis can activate inflammatory processes on the skin [37,38], such as bumps full of pus, acne scabbing, and scarring. Upon stress, one response is a corticotropin-releasing hormone, which elevates sebaceous lipogenesis [39]. In addition, a neuropeptide, i.e., substance P, is released from the peripheral nerves consequent to stress, which triggers the proliferation of sebaceous glands and lipid inception [38]. Our results are in accordance with the above pathogenic theories, showing moderate oily skin from students, in proportion with the prevalence of moderate stress. Likewise, two studies from Saudi Arabia [6] and Singapore [7] portrayed a significant association between stress level and acne severity.

Various stressors may elicit stereotype response. The current study, taking academics as a stressor, scrutinizes physical symptoms including physiological and mental factors (Table 3) of medical students. Due to stress, numerous physical symptoms may prompt and induce the secretion of catecholamines and cortisol (an anti-inflammatory hormone). Extended stress may perpetuate cortisol dysfunction, widespread inflammation, and pain. Studies have demonstrated altered cortisol levels as a response to pain and examination stress in medical students [40,41,42,43,44]. A majority of KKU medical students reported no physical symptoms like headache, tense muscles, and fatigue, although laboratory cortisol levels of participants were not tested in association with physical symptoms. However, the categorization of academic stress indicates that a majority of participants were moderately stressed, thus predicting no physical symptoms. Participants’ reports of headache, tense muscles, and fatigue may be due to improper posture, sedentary behavior, or a poor lifestyle, as also suggested by Bruflat et al. [45].

Studies conducted in Saudi Arabia reported sleep deprivation among one third of medical students [46,47]. Global data showed poor sleeping habits in 51–59% of medical students in United States and Lithuania [48,49,50]. In the race for better academic achievement, additional study hours and high workload strongly affect sleeping hours. Sleep plays a substantial role in cognitive skills, mental and physical health, academic performance, and psychiatric disorders [50,51,52,53,54,55,56,57]. Not many studies have been conducted to associate academic stress with sleep. However, a few studies have suggested poor sleeping patterns, depression, anxiety, and restlessness as comorbidities with stress [57,58]. Evidently, the result of our study is in agreement with the above studies showing a plurality of insomnia and self-reported psychiatric conditions (anxiety, phobias, depression, restlessness) with a significant positive association with overall academic stress. Based on these results, the study documents the impact of academic stress on mental and physiological factors.

Study limitation: Few males participated in this study, therefore the results cannot be generalized. In addition, there might be some biased answers provided for self-reported measures. In future, an equal number of male and female participants will be considered for research.

## 5. Conclusions

The study reports moderate stress in KKU medical students. It also demonstrates the significant positive correlation with acne and other physical symptoms. Highlighting these results, medical students should stay alert to the consequences of academic stress. It is recommended to medical colleges to organize counselling and programs to enhance academic skills among students and promote better management of stressful events, that can have ramifications for health.

## Figures and Tables

**Table 1 ijerph-19-08725-t001:** Demographic characteristics and distribution of the participants.

	*N* = 168	%
Male	19	11.3
Female	149	88.7
Age (Years)		
18 to 21	17	10.1
22 to 25	108	63.9
26 above	43	26.0
Height-cm (Mean ± SD)	158.2 ± 12.8	
Weight-Kg (Mean ± SD)	58.5 ± 14.5	
Place of study		
Abha	112	66.3
Khamis Mushayt	56	33.7
Marital status		
Single	120	71.4
Married	48	28.6
Department		
Medical	168	100
Study Level		
III	9	5.3
IV	7	4.2
V	6	3.6
VI	12	7.1
VII	3	1.8
VIII	19	11.3
IX	4	2.4
X	38	22.7
Internship	70	41.6

**Table 2 ijerph-19-08725-t002:** Comparison of academic stress score categories in participants.

	Low Academic Stress	Medium Academic Stress	High Academic Stress	*p* Value
	(*n* = 53, 31.54%)	(*n* = 98, 58.34%)	(*n* = 17, 10.12%)	
Score range	0–18	19–37	38–56	
Mean ± SD	8.55 ± 6.55 ^a^	26.71 ± 5.25 ^b^	45.46 ± 5.45 ^c^	<0.0001

Mean ± SD values of each score category differ at *p* < 0.0001. Values not sharing same superscripts differ significantly at 0.01 level of significance.

**Table 3 ijerph-19-08725-t003:** Comparison and correlation of overall academic stress with acne symptoms and other physical symptoms.

Academic Stress (Mean ± SD)	Acne Symptoms (Mean ± SD)	Other Physical Symptoms (Mean ± SD)	*p* Value
23 ± 12.3 ^a^	19 ± 7.6 ^b^	4.3 ± 3.3 ^c^	<0.01
Correlation of academic stress with acne symptoms and other physical stress symptoms			
Academic Stress	r = 0.20	r = 0.37	
	*p* = 0.0093 *	*p* = 0.00001 **	

Values not sharing same superscripts differ significantly at 0.01 level of significance. * *p* value < 0.01, ** *p* value < 0.00001.

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
