# Peer review of "Association of Academic Stress, Acne Symptoms and Other Physical Symptoms in Medical Students of King Khalid University"

_ijerph, 2022, doi:10.3390/ijerph19148725_

Round 1

Reviewer 1 Report

The paper is interesting, but there are too many unclear sentences, grammar mistakes and typos that make the paper not readily understood. The English needs improvement.

Several conclusions are faulty or are explained poorly so that their importance or validity are not understood.

Please find attached the paper with highlighted sentences that have grammar mistakes or other faults. Some comments are also given directly in the attachment.

Author Response

Reviewer comments are highlighted by yellow color.

All comments are fulfilled in the file attached.

Reviewer 2 Report

1. Introduction should be more elaborated: find more informations about other dermatological diseases that stress can intensify

2. Please add in the discussion

https://pubmed.ncbi.nlm.nih.gov/33212977/

2. I think article should have a point: study limitation: for example more female took part in the study, maybe in the future You should think to compare both male and female. Please think what other study limitations has this article (maybe season of making the survey?)

3.    188-190: Written informed consent for publication must be obtained from participating patients who 188 can be identified (including by the patients themselves). Please state “Written informed consent has 189 been obtained from the patient(s) to publish this paper” if applicable.

should it be in this place?

4. Did You ask if students had families, children? This can also intensify stress connecting with studying.

Author Response

Response to reviewer 2 is highlighted with pink.

  1. Introduction is elaborated by adding information - Line 33-36, 48-52.
  2. Reference added in discussion - Line 146-147.
  3. Study limitation is included -  Line 191-194.
  4. Removed "written informed consent" -Already mentioned in methodology section.
  5. Only marital status was asked.

Round 2

Reviewer 1 Report

The authors do not present a point-by-point answer to their changes and comment on text without including the reviewer's original comment. It is very difficult to see what and why is changed without reading both versions simultaneously, which is time-consuming and perhaps not optimal.

The English and the mistakes needing correction were highlighted by me to help the authors correct these. There has been no attempt to correct the English. The paper is still full of typos and grammar mistakes.

Author Response

All the reviewer comments are  fulfilled point to point and line by line, using comments feature. 

General questions from reviewer side are justified in using comments section.

All modifications are highlighted and assigned a comment number. 

English language is checked and corrected.

None of the comment left unanswered.
